META-RESEARCH

# Centralized scientific communities are less likely to generate replicable results

**Abstract** Concerns have been expressed about the robustness of experimental findings in several areas of science, but these matters have not been evaluated at scale. Here we identify a large sample of published drug-gene interaction claims curated in the Comparative Toxicogenomics Database (for example, benzo(a)pyrene decreases expression of SLC22A3) and evaluate these claims by connecting them with high-throughput experiments from the LINCS L1000 program. Our sample included 60,159 supporting findings and 4253 opposing findings about 51,292 drug-gene interaction claims in 3363 scientific articles. We show that claims reported in a single paper replicate 19.0% (95% confidence interval [CI], 16.9–21.2%) more frequently than expected, while claims reported in multiple papers replicate 45.5% (95% CI, 21.8–74.2%) more frequently than expected. We also analyze the subsample of interactions with two or more published findings (2493 claims; 6272 supporting findings; 339 opposing findings; 1282 research articles), and show that centralized scientific communities, which use similar methods and involve shared authors who contribute to many articles, propagate less replicable claims than decentralized communities, which use more diverse methods and contain more independent teams. Our findings suggest how policies that foster decentralized collaboration will increase the robustness of scientific findings in biomedical research.
DOI: https://doi.org/10.7554/eLife.43094.001

**VALENTIN DANCHEV\*, ANDREY RZHETSKY AND JAMES A EVANS\***

**\*For correspondence:** vdanchev@
stanford.edu (VD); jevans@
uchicago.edu (JAE)

**Competing interests:** The authors declare that no competing interests exist.

## Introduction

Concerns over reliability (*Ioannidis, 2005*) and reproducibility (*Prinz et al., 2011*; *Begley and Ellis, 2012*) in biomedical science call into question the cumulative process of building on prior published results. In a publication environment that rewards novel findings over verifications (*Nosek et al., 2015*; *Alberts et al., 2015*), the replicability of research claims that biomedical scientists assemble into biological models, drug development trials, and treatment regimes remains unknown (*Begley and Ellis, 2012*; *Yildirim et al., 2007*). Exact replications of biomedical research (*Errington et al., 2014*) occur only on small scales due to prohibitive expense and limited professional incentive.

Replication failures are typically attributed to systemic bias in a publication system that favors positive results (*Ioannidis, 2005*). This incentivizes questionable research choices such as p-hacking (*Head et al., 2015*; *Simonsohn et al., 2014*), 'flexible' data analysis (*Simmons et al., 2011*), low statistical power (*Dumas-Mallet et al., 2017*), selective reporting (the 'file drawer problem') (*Rosenthal, 1979*), and confirmation bias (*Nuzzo, 2015*). These questionable choices, combined with incomplete reporting of statistical methods and data (*Nosek et al., 2015*), contribute to the publication of false results that are unlikely to replicate in future experiments (*Simmons et al., 2011*).

Here we investigate the community that coalesces around a drug-gene interaction claim. We hypothesize that a decentralized community of largely independent, non-overlapping teams, which draws from a diverse pool of prior publications, using distinct methods under varying experimental conditions, is more likely to

produce robust results. Conversely, we expect that a centralized community involving repeated collaborations and using a narrow range of methods, knowledge from prior publications and experimental conditions is likely to produce less robust results. Unfortunately, repeated collaboration (*Hicks and Katz, 1996*; *Guimerà et al., 2005*), growing teams (*Wuchty et al., 2007*), star scientists (*Merton, 1968*; *Azoulay et al., 2014*), expensive shared equipment, and common citations (*Evans, 2008*; *Simkin and Roychowdhury, 2005*; *White et al., 2004*) are defining characteristics of the biomedical research enterprise today (*Hicks and Katz, 1996*; *Hand, 2010*).

Prior simulations have suggested that independent labs are less prone to peer pressure than a densely connected network of scientists, in which misleading early results can propagate more easily (*Zollman, 2007*; *Payette, 2012*). Related research on the 'wisdom of crowds' (*Lorenz et al., 2011*) and the exploration-exploitation trade-off (*Lazer and Friedman, 2007*) also found densely connected networks to be inefficient, and suggested that networks of semi-isolated subgroups would lead to an improvement in collective performance (*Fang et al., 2010*). A more recent experiment demonstrates that decentralized networks, rather than independence, may most improve collective performance. In estimation tasks completed by networks of individuals, it was found that the dominance of central individuals in networks tended to bias the collective estimation process and decrease the average accuracy of group estimates (*Becker et al., 2017*).

A separate body of literature attributes robustness of scientific findings to diverse methods (*Kaelin, 2017*; *Wimsatt, 2012*) used to corroborate them or distinct theories used to motivate them. A classic example is Jean Perrin's use of multiple experimental techniques and theories to precisely determine Avogadro's number (*Salmon, 1984*). Nevertheless, there has been no comprehensive evaluation of the relationship between the way scientific communities are networked and the robustness and replicability of published findings. Moreover, when empirical data on scientific collaboration have been used (*Guimerà et al., 2005*), the outcomes of collective performance have typically been measured indirectly (e.g., via article or journal citations). Similarly, literature on research

reliability has focused on methodological features rather than the way scientific communities are networked. Moreover, there has been relatively little research of this nature in the field of biomedical science.

Here we demonstrate a strategy for evaluating the replication likelihood for tens of thousands of drug-gene interaction claims. This strategy builds on the synergy of two advances. First, databases of empirical claims curated from the scientific literature in certain subject areas such as molecular toxicology and biochemistry (*Davis et al., 2017*) can be linked to databases of scientific articles such as MEDLINE and the Web of Science to systematically analyze features that characterize the provenance of a scientific claim (*Evans and Foster, 2011*) such as authors, affiliations, and the number of experiments for and against. Second, data from high-throughput experiments (*Subramanian et al., 2017*) performed by robots allow researchers to estimate the replication likelihood for many published claims. Here we report the results of analyses performed on claims identified by comparing the Comparative Toxicogenomics Database (CTD; *Davis et al., 2017*) and the LINCS L1000 experiment (*Subramanian et al., 2017*; see *Figure 1* and Materials and methods).

The CTD had recorded over 1.26 million drug-gene interaction (DGI) claims published in journals as of June 7, 2016, with each claim being a 'triple' of a drug, a gene and an interaction effect. We selected interaction effects in which a drug either 'increases expression' or 'decreases expression' of an mRNA in human tissues. This resulted in a sample of 239,713 DGI claims curated from 11,754 scientific articles. The LINCS L1000 experiment generated 1.3 million gene expression profiles from 42,080 perturbagens across a range of different cell lines, time points and dosage levels. Each profile consisted of a drug, a gene (mRNA) and a z-score. The LINCS L1000 experiment consolidated multiple expression profiles to generate a moderated z-score for each experimental condition, and we combined these into a single combined z-score for each drug and gene (mRNA). We matched these triples from the LINCS L1000 experiments to triples in the CTD, and found 51,292 drug-gene interactions at the intersection, corresponding to 60,159 supportive findings and 4253 opposing findings from the literature, annotated from 3363 scientific articles

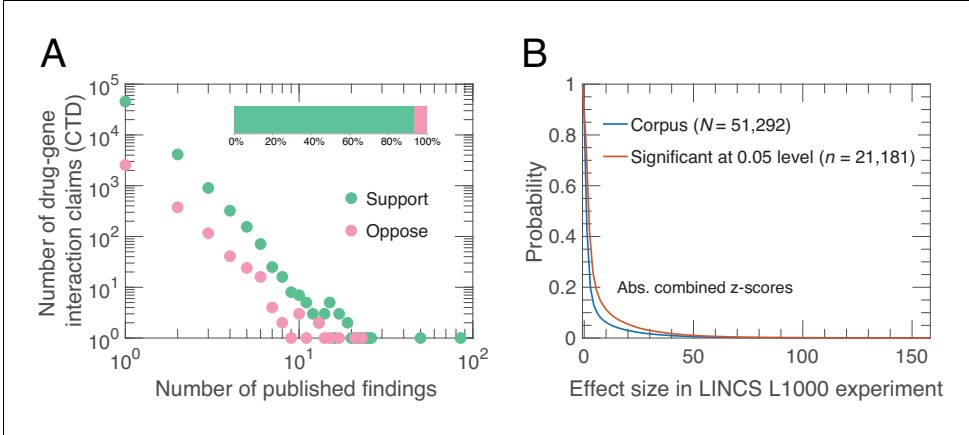

**Figure 1.** Alignment of drug-gene interaction (DGI) claims reported in the literature with DGI claims from high-throughput experiments. (A) Our analysis of 51,292 DGI claims (see *Supplementary file 1*) in the literature revealed 60,159 supporting findings (green) and 4253 opposing findings (pink) in aggregate. These DGI claims co-occurred in both the CTD publication dataset and the LINCS L1000 exerimental dataset. Most claims (45,624) were supported by just one published finding, 4127 claims were supported by two published findings, and the remaining 1541 claims were supported by three or more published findings. Some claims (3154) were both supported and opposed by the published findings, meaning that in addition to the supporting finding(s), there is one or more increase/decrease interactions in the CTD dataset that propose the opposite effect: 2563 claims were opposed by one, 376 by two, and 215 by three or more published findings. Please note that both axes in the main graph are logarithmic. (B) We calculated experimental effect sizes with combined $z$-scores for the 51,292 DGI triples in the LINCS L1000 dataset. This graph plots the probability (y-axis) versus absolute value of the combined $z$-score for all triples (blue line) and those that are significant at the 0.05 level (salmon line). Significant in this context means that the drug-gene effect is observed across a range of experiment conditions; the method used to determine significance is described in Materials and methods.

DOI: https://doi.org/10.7554/eLife.43094.002

The following figure supplements are available for figure 1:

**Figure supplement 1.** Publications that share authors are more likely to agree about the direction of a drug-gene interaction than publications with distinct authors, computed among pairs of papers reporting claims in the sub-corpus of 2493 claims.

DOI: https://doi.org/10.7554/eLife.43094.003

**Figure supplement 2.** Estimates of probability density functions for variables of interest in our corpus using a normal kernel function.

DOI: https://doi.org/10.7554/eLife.43094.004

(see *Supplementary file 1*). We use this sample to estimate how the probability of claim replication depends on support in the literature, social independence, methodological independence, knowledge independence, scientist centralization, journal prominence, and experimental variability (see Materials and methods).

Our high-throughput replication strategy evaluates the replicability of a broad scientific claim rather than the reproducibility of any single experiment contributing to that claim. (The evaluation of an individual experiment requires the original experimental protocol to be repeated in the new experiment [*Errington et al., 2014*].) Nevertheless, collective agreement across many published findings provides evidence for a robust claim – that is, a claim that withstands changes in the technique used (*Wimsatt, 2012*; *Nosek and Errington, 2017*; *Kaelin, 2017*), the scientists doing the research and the experimental setting. Such claims form a solid foundation for further research and, potentially clinical trials based on the claim.

## Results

### Distribution and agreement of experiments

We observe a long-tailed distribution of published findings in support of a given DGI claim (*Figure 1A*). Most claims are supported by findings in one (89%) or two (8%) articles, while few appear in many articles. There is wide consensus in the literature: the vast majority of published

findings (93%) agree on the direction of DGI claims. Among the 11% of claims supported by two or more published findings, the agreement increases to 94%. In contrast, only 41% (21,181/51,292) of the DGI effect-sizes in LINCS L1000 generalize across experimental conditions (*Figure 1B*). Although those two quantities – agreement among published findings and generalizability of LINCS L1000 effect sizes – are not directly comparable as they utilize different measurements, the overwhelming agreement observed in the literature compared to the LINCS L1000 data suggests that the literature may influence the selection and interpretation of experimental results by scientists through confirmation bias. One consequence of this is that experimental results that contradict preexisting knowledge are 'filed away' rather than submitted to journals (*Rosenthal, 1979*).

Further, we find that consensus in the biomedical literature is strongly and positively associated with connections between papers arising from overlapping authors. A pair of findings about a DGI claim reported in papers with some of the same authors are significantly more likely to agree (0.989, 2486 of 2514 overlapping pairs of papers) than findings reported in papers by socially independent teams (0.889, 18,527 of 20,846 non-overlapping pairs of papers), with a mean difference of 0.1001 (95% CI, 0.094–0.106, $P < .0001$, 100,000 bootstrap iterations; *Figure 1—figure supplement 1*).

### Replication rates

For a given set of claims it is possible to estimate a random or baseline replication rate $RR_{rand}$ and an observed replication rate $RR_{obs}$ (along with 95% confidence limits for both) using the approach outlined in 'Measuring relative replication increase' (see Methods and materials). For our sample we estimate $RR_{obs}$ to be 0.556 (95% CI, 0.553–0.558, $N = 51,292$ claims) and $RR_{rand}$ to be 0.503 (95% CI, 0.498–0.506, $N = 51,292$ claims): this corresponds to a percentage relative replication increase ($RRI = ((RR_{obs} - RR_{rand})/RR_{rand}) \times 100\%$) of 10.6% (95% CI, 9.3%–11.8%). Figure 2D shows that, as expected, DGIs that generalize across experimental conditions in LINCS L1000 are more likely to replicate published DGI claims ($RR_{obs} = 0.604$, 95% CI, 0.597–0.610, $n = 21,181$ claims) than DGIs that do not generalize ($RR_{obs} = 0.522$, 95% CI, 0.516–0.528, $n = 30,111$ claims). Indeed, the replication rate for the latter group is only marginally higher than $RR_{rand}$ for this group ($RR_{rand} = 0.501$, 95% CI, 0.496–

0.507, $n = 30,111$ claims). Encouragingly, this suggests that some disagreement within the literature is attributable to experimental and biological variation in the experiments performed by different scientists. In the subsequent analysis, we consider generalized LINCS L1000 DGIs because only those can serve to evaluate the replicability of published claims.

### Collective correction in science

A central concern is whether the replication problem applies only to novel and rare claims or if it also afflicts widely supported results, as recently hypothesized (*Nissen et al., 2016*; *McElreath and Smaldino, 2015*). To examine this question, we integrated collective findings using a binomial Bayesian model (*Gelman et al., 2013*) with a uniform prior that accommodates skewed distributions (*Davidson-Pilon, 2015*) like that of findings per claim we observed. The model allocates higher probability to scientific claims unanimously supported in a large number of articles and lower probability to infrequent and disputed claims (*Figure 2A*). The resulting posterior distributions of support were used to categorize DGI claims into five classes of support: Very High, High, Moderate, Low support, and Not Supported (see *Figure 2B* and Materials and methods).

*Figure 2E* shows that claims with Very High support in the biomedical literature ($RRI = 45.5\%$; 95% CI, 21.8–74.2%) with an average of 6.9 papers confirming the claim, and claims with High support ($RRI = 34.5\%$; 20.2–50.3%) with an average of 3.3 confirming papers, are substantially more likely to replicate in high-throughput experiments than those with Low and Moderate support ($RRI = 19.0\%$; 16.9–21.2% and 16.2%; 9.8–22.9%, respectively). The replication of claims with Low and Moderate support is consistent with reproducibility estimates reported in the literature, ranging from 11% ($N = 67$; *Begley and Ellis, 2012*) to 25% ($N = 53$; *Prinz et al., 2011*). Claims with Very High and High support replicate at a much higher rate, whereas Not Supported claims are significantly less likely to replicate than random ($RRI = -28.9\%$; $-61.9\%$–16.7%). They are also associated with greater experimental variability (*Figure 2F*), confirming that collective disagreements among findings truthfully signal experimentally unstable interactions peculiar to specific contexts and unlikely to replicate.

Logistic models adjusting for experimental variability confirm the positive relationship between scientific support in the literature $L_{supt}$

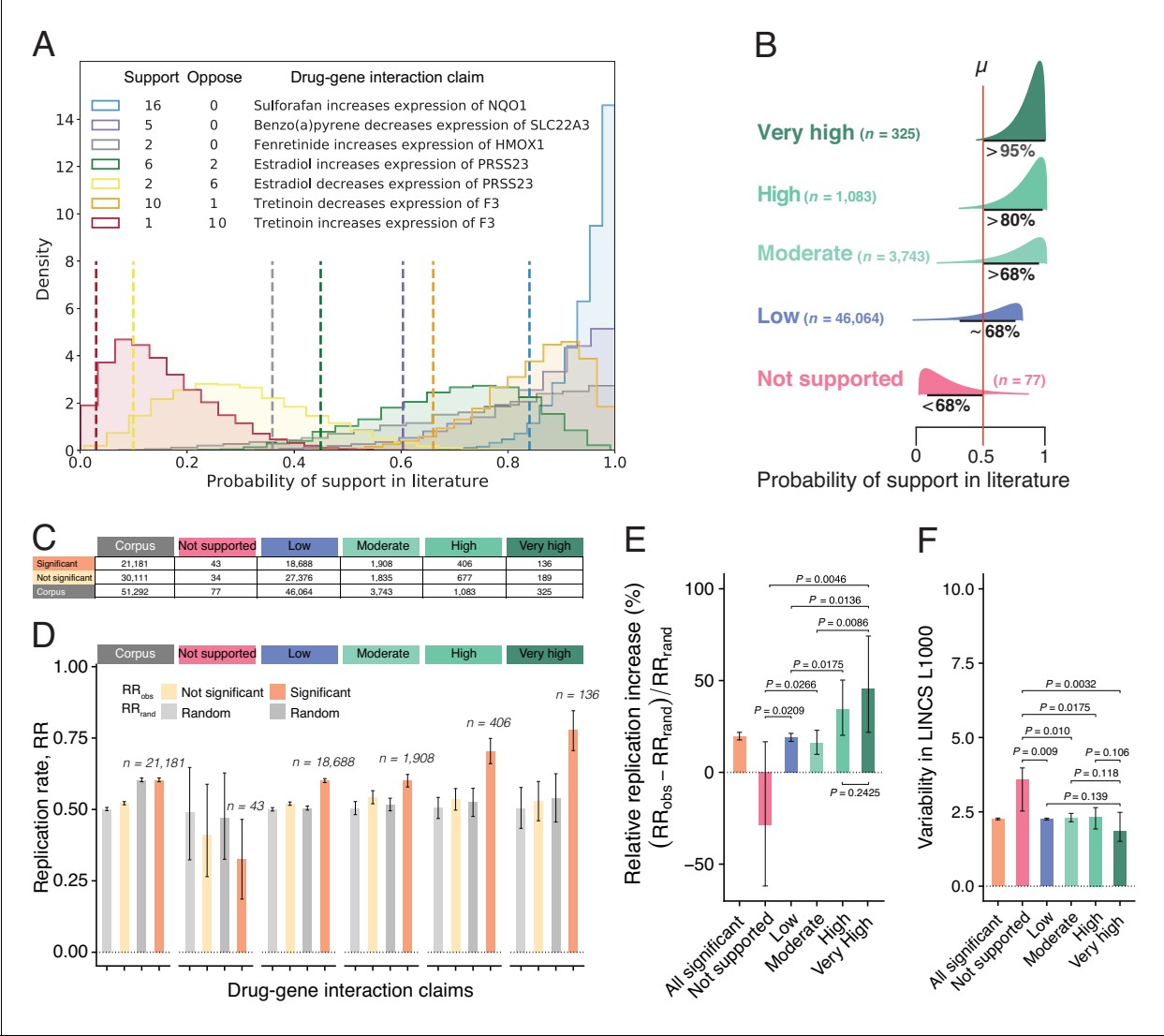

**Figure 2.** Estimates of claim replication as a function of the probability of support in the literature and generalizability across high-throughput experiments. (A) Posterior distributions of probability of support in the biomedical literature for a sample of seven DGI claims for which there are at least two findings (supporting and/or opposing). Note that the top three claims receive only supporting findings in the literature, whereas the fourth and fifth claims are opposites (so papers that support the fourth claim oppose the fifth claim, and vice versa), and likewise for the sixth and seventh claims. We obtained model estimates for each claim by performing 10,000 Markov chain Monte Carlo (MCMC) sampling iterations (see Materials and methods). For each claim, we summarize the probability of support (dashed vertical line) using the lower bound on the one-sided 95% posterior credible interval: this value ranges from 0.84 for a claim that is supported by 16 findings and opposed by no findings, to 0.03 for a claim that is supported by one finding and opposed by 10 findings. (B) DGI claims in the literature can be categorized into one of five classes of support (Very High; High; Moderate; Low; Not Supported) on the basis of distributions like those in panel A; the number of claims included in each class is shown in brackets. (C) Number of DGI claims that are significant (second row) and not significant (third row) at the 0.05 level in the LINCS L1000 dataset for the whole corpus (second column) and for each of the five classes of support in the literature (columns 3–7). (D) Observed replication rates ($RR_{obs}$) and expected replication rates ($RR_{rand}$) for claims that are significant and non-significant in the LINCS L1000 dataset for the whole corpus (left) and for each of the five classes of support in the literature. (E) The relative replication increase rate ($RRI = 100 \times \frac{RR_{obs} - RR_{rand}}{RR_{rand}}$) for claims that are significant in the LINCS L1000 dataset (left) and for each of the five classes of support in the literature. (F) Variability (coefficient of variation) in the LINCS L1000 dataset across cell lines, durations and dosages for claims that are significant in this dataset (left) and for each of the five classes of support in the literature. Statistical significance and error bars were determined by bootstrapping (see Materials and methods). All error bars represent 95% CI.

DOI: https://doi.org/10.7554/eLife.43094.005

The following figure supplements are available for figure 2:

**Figure supplement 1.** Replication increases with claim's probability of support in the literature.

DOI: https://doi.org/10.7554/eLife.43094.007

*Figure 2 continued on next page*

*Figure 2 continued*

**Figure supplement 2.** Description of claim types in the whole corpus of 51,292 claims.

DOI: https://doi.org/10.7554/eLife.43094.006

and the probability of replication success (*Figure 2—figure supplement 1*). These results suggest that findings reported in a single biomedical article are likely fallible. Viewed as a complex system of converging and diverging findings, however, biomedicine exhibits collective correction.

We note that the process of collective correction applies to a small subset of claims as the majority of claims (89%) in our corpus are only reported in a single paper. Multiple factors could account for why a large proportion of claims in the corpus are not reported in repeat experiments. The lower replication rate of single-study claims indicates that many of those novel claims were likely obtained by chance. Even if tested further, they are less likely to produce positive results compared to multiple-studied claims, and thus more likely to be filed away than submitted to a journal. This interpretation is supported by our finding that single-study claims in our corpus (ie, claims in the Low support class) have a $RR_{obs}$ of only 0.601, while the first published studies about multiple-studied claims that eventually achieve High or Very High support have a $RR_{obs}$ of 0.720.

### Networked scientific communities and replicability

Independent and decentralized sources of evidence should increase claim robustness. We examine the impact of social and technical dependencies on the replication of biomedical claims by fitting logistic regression models to predict replication success against our network dependency measures for each claim. We performed this analysis using subsamples of DGI claims that simultaneously: 1) received support from multiple papers in the literature (i.e., claims with Moderate and above support in the literature), thereby converging on an effect direction; and 2) generalized across conditions in LINCS L1000. The resulting subsample consists of 2493 claims, associated with 6272 supporting and 339 opposing findings from 1282 research articles (see *Supplementary file 2*). Despite the smaller size of this subsample, our analysis represents the largest biomedical replication of its kind to date. We restrict our analysis of dependencies to a subsample of published DGI claims

supported by multiple papers because single-paper claims cannot, by definition, exhibit network dependencies or centralization. By examining only interactions having significant agreement within both the literature and LINCS L1000, we can directly examine the effect of social, methodological, and knowledge dependencies on the replicability of published claims in high-throughput experiments (see *Figure 3* and *Figure 4*).

*Figure 4A* shows that the odds ratios (OR) of replication increase substantially with support in the literature $L_{supt}$ (OR 23.20, 95% CI, 9.08–59.3), social independence $S_{ind}$ (OR 6.31, 95% CI, 4.07–9.79), methodological independence $M_{ind}$ (OR 6.30, 95% CI, 3.44–11.53), and knowledge independence $K_{ind}$ (OR 5.53; 95% CI, 2.58–11.84). Consistent with this pattern, claim replication decreases sharply with centralization $C$ (OR 0.36, 95% CI, 0.27–0.48). Our estimates indicate that claim robustness, defined here as repeated instances of confirmatory decentralized evidence, increases replication success (see also *Figure 4—figure supplement 1*). When all predictors are modeled simultaneously (*Figure 4A*), centralization and support in the literature largely account for all of the others. This suggests that centralized and extensive biomedical collaboration is associated with use of the same biomedical research techniques and attention to the same prior research (see *Figure 3—figure supplement 1*).

Social dependencies could impact the composition of researchers studying a claim by discouraging researchers outside the dense communities of researchers who initially lay claim to it from pursuing or reporting their findings, as suggested in previous research (*Azoulay et al., 2015*). Focusing on High and Very High support claims with some degree of centralization (n = 295), we found that claims originally reported and subsequently confirmed by papers with overlapping authors (n = 117) resulted in much more centralized communities (mean $C$ = 0.6164; 95% CI, 0.5925–0.6402) compared to claims reported and subsequently confirmed by papers with independent authors (n = 178; mean $C$ = 0.4004; 95% CI, 0.3765–0.4242; two-way ANOVA test with unbalanced design.) This exploratory result suggests that claims

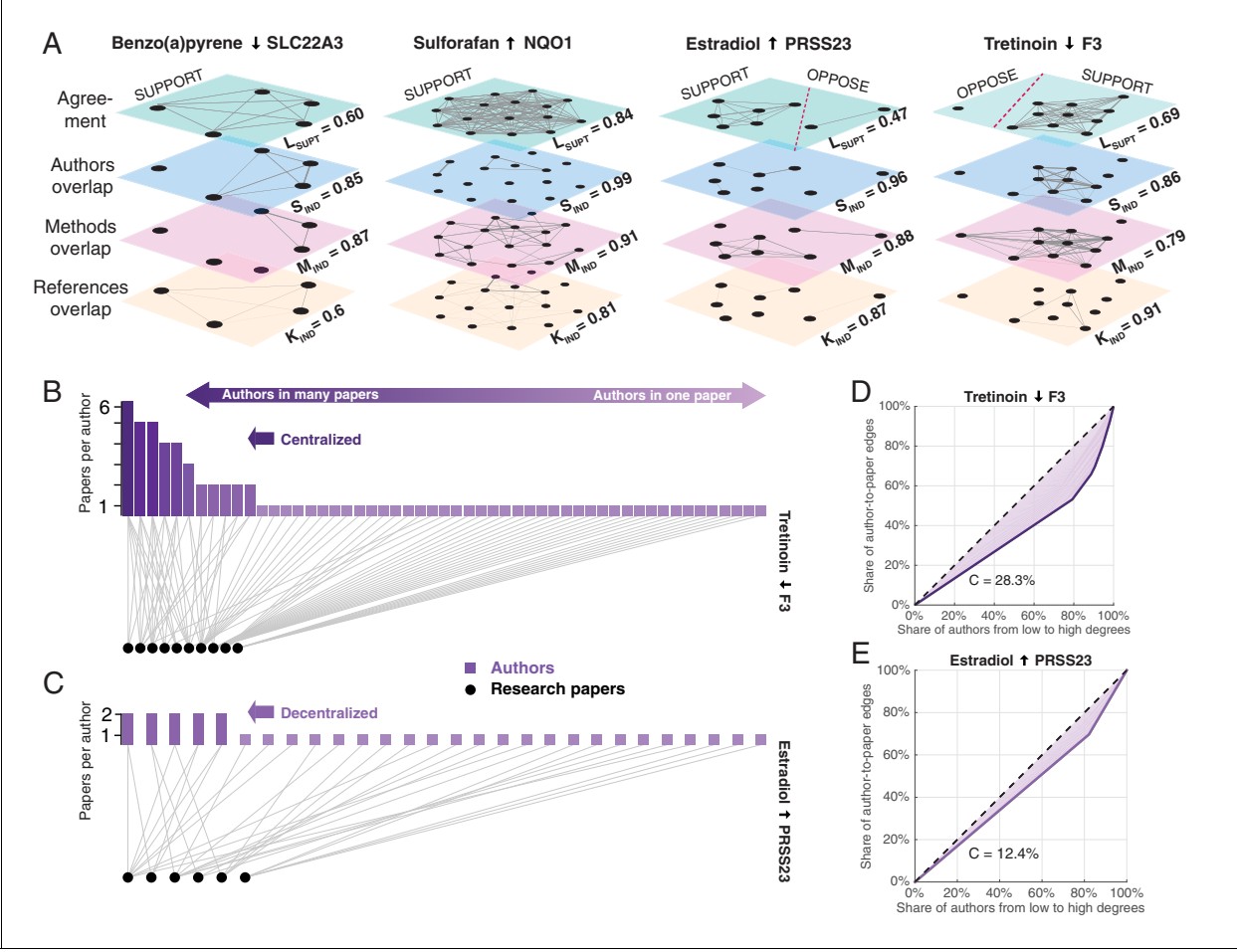

**Figure 3.** Exemplary networks comprising social, methodological, and references dependences and centralization patterns in scientific communities. (**A**) Multilayer networks for four of the claims shown in *Figure 2A*. The nodes in each layer are scientific papers. Pairs of papers are connected by an unweighted edge in the top layer if they agree on the effect direction, and by a weighted edge in the other layers if there is an overlap of authors (second layer), methodologies (third layer) or references to prior publications (fourth layer): the thickness of the weighted edges is proportional to the overlap (Jaccard coefficient; *JC*); for clarity, we only plot edges above the mean *JC* value in the third layer. Dashed red lines in the top layer separate supporting and opposing findings. Each layer is associated with a score: support in the literature $L_{supt}$, social independence $S_{ind}$, methodological independence $M_{ind}$, and knowledge independence $K_{ind}$ (see 'Network dependencies and centralization' in Methods and materials). Figures plotted with Pymnet (*Kivelä, 2017*). (**B**) Bipartite network with edges connecting authors (rectangles) to the papers they published (circles) for the 10 papers that support the claim shown in the fourth panel of *Figure 3A*. A small group of investigators author most of these papers, while most investigators author only one paper, making this a centralized network. The Gini coefficient (see Materials and methods) for this network is 28.3%. (**C**) Bipartite network for the six papers that support the claim shown in the third panel of *Figure 3A*. Here all investigators author relatively comparable numbers of papers: this decentralized network has a Gini coefficient of 12.4%. (**D, E**) Lorenz curves for the examples shown in B and C.

DOI: https://doi.org/10.7554/eLife.43094.008

The following figure supplements are available for figure 3:

**Figure supplement 1.** Pearson correlation coefficients between network indices.
DOI: https://doi.org/10.7554/eLife.43094.009

**Figure supplement 2.** Papers and pairs of papers are differentiated by the number of findings they report (in the sub-corpus of 2493 claims).
DOI: https://doi.org/10.7554/eLife.43094.010

exhibiting social dependencies very early in their development likely receive limited attention and reportage from researchers outside the communities that originated them.

An alternative explanation for replication success is the biological tendency for some DGI claims to generalize across conditions and so replicate in future experiments. *Figure 4A* shows that experimental variability has a negative, marginally significant effect on replication, but support from multiple teams organized in

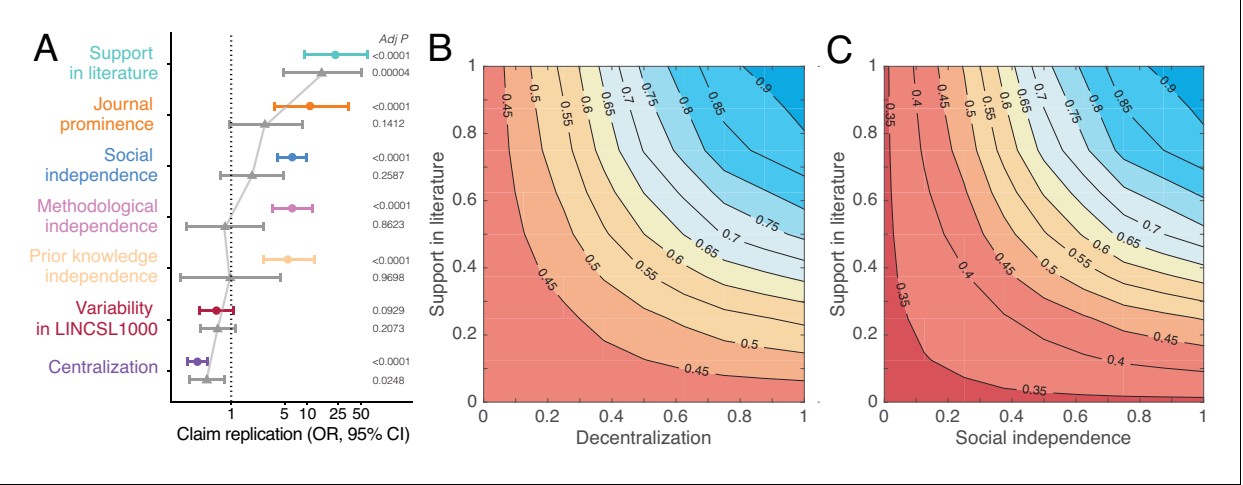

**Figure 4. Predictors of replication success.** (A) Odds ratios derived from logistic regression models with claim replication as the response variable and seven predictors modeled independently (disconnected colored dots; n = 2493) and simultaneously (connected grey triangles; n = 2491). Predictors are rescaled $\frac{x_i - \min(x)}{\max(x) - \min(x)}$ for comparability. P-values are adjusted for multiple comparisons using the Benjamini–Hochberg (*Benjamini and Hochberg, 1995*) procedure. (B–C) Contour plots showing that the predicted probability of claim replication (derived from logistic regression models with interaction terms, see also *Figure 4—figure supplements 2,3*) increases with decentralization and support in the literature (B), and with social independence and support in the literature (C), after adjusting for variability in LINCS L1000.

DOI: https://doi.org/10.7554/eLife.43094.011

The following figure supplements are available for figure 4:

**Figure supplement 1.** Predictors of replication success.

DOI: https://doi.org/10.7554/eLife.43094.012

**Figure supplement 2.** Claims reported by centralized communities less likely replicate.

DOI: https://doi.org/10.7554/eLife.43094.014

**Figure supplement 3.** Claims reported by multiple socially independent teams more likely replicate.

DOI: https://doi.org/10.7554/eLife.43094.015

**Figure supplement 4.** Estimates of probability density functions for our variables on the sub-corpus of claims with determined direction of the drug-gene effect in CTD and LINCS L1000.

DOI: https://doi.org/10.7554/eLife.43094.013

**Figure supplement 5.** Support in the literature and decentralization of scientific communities remain strong and significant predictors of claim replication success after we account for multicollinearity.

DOI: https://doi.org/10.7554/eLife.43094.016

decentralized communities are much more informative predictors of replication success.

Our combined model also accounts for journal prominence $J$, which we measure with journal eigenfactor (*Bergstrom et al., 2008*), a score that credits journals receiving many citations from highly cited journals. Claim replication increases with journal prominence (*Figure 4A* and *Figure 4—figure supplement 1*), but prominent journals are responsible for only a tiny fraction of all claims. This warrants our evaluation strategy and the practice of extracting and archiving findings from a wide range of journals (*Davis et al., 2017*).

*Figure 4B* shows that by accounting for biomedical support and decentralization, we can identify claims with high predicted probability of replication. Claims supported by many

publications have about 45% higher predicted probability to replicate when investigated by decentralized versus centralized communities. Even if a DGI claim garners wide support, if it is studied exclusively by a centralized scientific community, the claim has a predicted probability of replication that is similar to that for a claim reported in a single paper. It is unlikely that such a claim will clear a clinical trial or enter medical practice (see *Figure 4—figure supplement 2*). This suggests that collective correction in science can be undermined when one or several scientists exercise disproportionate influence on research across multiple investigations of a claim. Likewise, claims robust to many, socially independent investigations have 55% higher predicted probability of replication than those studied by a few overlapping collaborations

(*Figure 4C* and *Figure 4—figure supplement 3*). All models adjust for experimental variability in the LINCS L1000 data. Results are robust to outliers and multicollinearity (see Appendix 1).

## Discussion

This paper repurposes high-throughput experiments to evaluate the replicability of tens of thousands of DGI claims from the biomedical literature. It provides evidence that research replicability is associated with the way scientific collaborations are networked, after adjusting for variability in high-throughput experiments and the methodology of published studies. Centralized, overlapping communities with shared methodologies and exposure to the same prior knowledge are supported by extensive collaboration (*Hicks and Katz, 1996*; *Hand, 2010*), ubiquitous communication technologies, a reward system that places a premium on productivity, and cumulative advantage processes that create central, star biomedical scientists (*Azoulay et al., 2014*). Our results indicate that such communities are associated with scientific findings of lower replicability. Decentralized, sparsely connected communities explore diverse research methodologies and prior knowledge (*Figure 3—figure supplement 1*) and are more likely to generate replicable findings.

Our work addresses widely shared concerns about the robustness of research results in biomedicine. Recent work (*Nissen et al., 2016*; *McElreath and Smaldino, 2015*) submitted that reliability issues uniformly impact scientific findings, afflicting even widely accepted 'facts'. Nissen et al. reason that even if many studies converge on the same finding, this may not increase our confidence because the same systematic biases – publication bias in particular – that made the original finding false are also likely to affect subsequent findings, resulting in a canonization of 'false facts' (*Nissen et al., 2016*). Our analysis identifies conditions under which such an argument holds, namely for research claims studied by centralized, overlapping collaborations. In the absence of social independence, replicability is low and the likelihood of a claim being revised or withdrawn is virtually non-existent as authors and methods reinforce agreement with themselves. Both disagreement and replication increase when teams from a decentralized or sparsely connected community provide separate confirming evidence for a claim. Our findings allay science policy concerns over a universal 'replication crisis' and

identify conditions – decentralized and novel collaborations – that facilitate collective convergence on replicable findings in biomedical science.

Our findings highlight the importance of science policies that promote decentralized and non-repeated collaborations. We acknowledge that certain 'big science' initiatives (*Hand, 2010*), such as the human genome project (*International Human Genome Sequencing Consortium, 2001*), involve large international consortiums that require a degree of repeated collaboration and centralization by design. It is also the case that the current organization of science incentivizes repeated collaborations (*Guimerà et al., 2005*; *Lungeanu and Contractor, 2015*; *Hilton and Cooke, 2015*), including centralized communities revolving around star scientists (*Azoulay et al., 2014*) or prestigious and well-endowed institutes. By reducing information and coordination costs, repeated collaborations are more productive than new ones (*Hilton and Cooke, 2015*). Consequently, projects that involve prior collaborators are more likely to be funded (*Lungeanu et al., 2014*) and successfully completed (*Cummings and Kiesler, 2008*). Such positive feedback, however, can lead to the lock-in of rigid collaborative clusters, which produce voluminous scientific output with diminished value for the wider biomedical field. Science policies supporting biomedicine should account for the trade-off between increased productivity and diminished reliability.

Our choice of repurposing the LINCS L1000 data to estimate the replication likelihood of published claims places importance on unfiltered high-throughput experiments for our results. We note, however, that our approach does not rely on the LINCS L1000 data being completely free from error. Rather, we argue that the LINCS L1000 data are: i) produced with methods representative of DGI research; ii) free from social, prior knowledge and narrow methodological dependencies associated with the sequential publication of findings by communities of researchers. In this way, our manuscript attempts to elucidate biases introduced by the social, cultural and methodological structure of science to noisy experimental data. From this perspective, LINCS L1000 experiments must remain unfiltered by the publication system and the expectations of any particular scientific community. In the current state of the biomedical literature, where most reported results are confirmatory, scalable approaches for identifying uncertain claims are in short supply. Experimental data

such as the LINCS L1000 provides an informed approach to evaluate published claims.

This paper demonstrates an approach to evaluate the replicability of potentially vast numbers of published biomedical claims simultaneously. With the proliferation of high-throughput experimental platforms and improvements in cross-platform reproducibility (*Haibe-Kains et al., 2013*), the approach we report here could be further refined and extended to a continuous replication system (*Goodman et al., 2016*) that revises scientific results in light of new experimental evidence, increasing the credibility and robustness of biomedical knowledge.

## Materials and methods

### High throughput claim replication strategy

We examined a corpus of 51,292 scientific claims about drug-gene interactions in human systems. We compiled the corpus by using claims about directed drug-gene interactions (DGIs) curated from biomedical publications in the Comparative Toxicogenomics Database (CTD) (*Davis et al., 2017*). Each scientific claim is a triple of drug, gene, and interaction effect. For comparability with high-throughput experiments, we selected interaction effects in which a drug 'increases expression' or 'decreases expression' of an mRNA in humans (effect magnitudes were not recorded), amounting to 239,713 DGI claims curated from 11,754 scientific articles. The CTD provides PubMed IDs of articles in which the finding is reported, which enabled examination of article content (e.g., methods) and metadata (e.g., authors, citations).

To estimate replication likelihood, we map our corpus of DGI claims to high-throughput experimental data from the NIH LINCS L1000 program, which was performed at the Broad Institute of MIT and Harvard. This program generated 1.3M gene expression profiles from 42,080 chemical and genetic perturbagens across cell lines, time points, and dosage levels (*Subramanian et al., 2017*). We used profiles induced by chemical perturbagens, amounting to 19,811 small molecule compounds (including FDA approved drugs). The LINCS L1000 data have been reported to be highly reproducible (*Subramanian et al., 2017*) compared to drug screen data generated via RNA sequencing, instances of which have been found to exhibit inconsistencies across platforms (*Haibe-Kains et al., 2013*).

The LINCS L1000 team consolidated multiple expression profiles or replicates into signatures corresponding to moderated (*Subramanian et al., 2017*) z-scores (Level five data in LINCS L1000). Each signature and corresponding moderated z-score is a representation of gene responses to drug perturbations under a particular cell line, dosage, and duration. We combined moderated z-scores for each DGI using a bootstrapped modification of Stouffer's method (*Whitlock, 2005*; *Himmelstein et al., 2017*) $Z = \frac{\sum_{i=1}^{k} z_i}{\sqrt{k}}$, where $z_i$ is a moderated z-score and $k$ is the number of moderated z-scores for the DGI. We bootstrapped (10,000 iterations) each sample of moderated z-scores per DGI to estimate confidence intervals. The samples vary across DGIs as a function of the number of cell lines, dosages, and durations under which the DGI was tested (the mean and median of moderated z-scores per DGI in our corpus are 143.8 and 49, respectively).

The above procedure generates triples of (i) drug, (ii) gene (mRNA), and (iii) combined z-score indicating experimental effect size and direction. We matched DGI triples from the LINCS L1000 experiments to DGI triples in CTD, and found 51,292 DGIs at the intersection, corresponding to 60,159 supportive and 4253 opposing findings from the literature, annotated from 3363 scientific articles (Appendix 1 details data sources). To verify that published findings have been obtained independently from the LINCS L1000 high-throughput data, a search in PubMed was performed using the search terms LINCS L1000 and L1000. The search identified no reference to the LINCS L1000 data among the 3363 articles.

### Experimental generalizability and variability

We used the confidence intervals for the combined z-scores we estimated via bootstrapping for each of the 51,292 DGIs to differentiate generalized from context-specific interactions in LINCS L1000. We classified DGIs as 'generalized' those significant at the 0.05 level (i.e., the corresponding 95% confidence intervals do not contain the null value of 0; see *Figure 2C and 2D*). For generalized/significant DGIs, we further examine robustness to experimental conditions (*Kaelin, 2017*; *Goodman et al., 2016*; *Van Bavel et al., 2016*) by measuring variability of each interaction in LINCS L1000 across cell lines, dosages, and durations using the coefficient of variation (*CV*). For a set of z-scores

about a DGI in LINCS L1000, $CV_z$ is defined as the ratio of the standard deviation to the absolute value of the mean $CV_z = \frac{\sigma_z}{\text{abs}(\mu_z)}$. $CV_z$ is a normalized measure of variability that allows us to make comparisons across DGIs.

## Bayesian model of scientific support

Claims about DGIs receive different proportions of supporting and opposing published findings. To estimate the probability of support or confirmation in the literature $L_{\text{supt}}$ for each DGI claim, we design a simple Bayesian model (*Gelman et al., 2013*; *Davidson-Pilon, 2015*; *Kruschke, 2014*). We assume that the prior distribution of θ is uniform on the interval [0,1]: $\theta_i$ ~Uniform(min = 0, max = 1). Further, we assume that the number of supporting published findings γ in *n* findings about that claim is drawn from a binomial distribution, p(γ|θ) ~ Bin(γ|n, θ). We approximated the posterior density of θ for each drug-gene claim by performing 10,000 Markov chain Monte Carlo (MCMC) sampling iterations (2,500 burn-in iterations) for each drug-gene claim using the Metropolis–Hastings MCMC sampler implemented in the PyMC package (version 2.3.6) for Python. To improve convergence, we approximate the maximum posterior (MAP) before running the MCMC sampler (*Davidson-Pilon, 2015*).

We used the posterior distributions from our Bayesian model of support to categorize DGI claims into classes. For each claim, we estimated the overlap between the posterior credible intervals (PCI) and the null value of μ = 0.5 (*Figure 2B*): Very High support claims (95% PCI exceeds μ) yield agreement from multiple papers (~7 papers on average), amounting to 325 claims supported by 2241 findings, but only opposed by 21; High support claims (80% PCI exceeds μ) yield agreement from 3 papers on average, comprising 1083 claims, supported by 3525 and opposed by 42 findings; Moderate support claims (68% PCI exceeds μ) yield agreement from 2 papers on average, comprising 3743 claims, supported by 7557 and opposed by 38 findings; Low support claims (68% PCI contains μ) are overwhelmingly supported by a single finding or opposed by virtually the same number of findings that support them such that the direction of the effect is undetermined, comprising 46,064 claims, supported by 46,735 and opposed by 3668 findings. Not Supported claims (68% PCI is smaller than μ) generate lower support than expected as a greater number of

papers reported findings in the opposite direction, comprising only 77 claims, supported by 101 and opposed by 484 findings (*Figure 2—figure supplement 2*).

## Measuring relative replication increase

A DGI is replicated, $R = 1$, if the direction of the effect size (i.e., positive or negative combined z-score) in LINCS L1000 matches the direction of the effect (i.e., increase or decrease) claimed in literature, and $R = 0$, otherwise. For the entire corpus and for selected subsets of the corpus, we created replication vectors [1, 0] and calculated observed replication rates $RR_{\text{obs}}$ by dividing the number of replicated claims by the total number of claims. We then bootstrapped (100,000 iterations) the replication vectors to estimate the 95% percentile confidence intervals of $RR_{\text{obs}}$ (*Figure 2D*). We empirically estimated the baseline or random replication rate $RR_{\text{rand}}$ (the proportion of random matches in which LINCS L1000 effects matched the direction of the effects reported in the literature) for the entire corpus and for various subsets of the corpus by iteratively matching published DGI claims to randomized combined z-scores in LINCS L1000 (100,000 random permutations). We then used the resulting permutation distributions to determine the 95% confidence intervals of $RR_{\text{rand}}$. The empirical baseline model corrects for unbalanced data, which would occur if more claims of a certain direction, either 'increasing' or 'decreasing', are present in both literature and high-throughput experiments. The percentage relative replication increase *RRI* is defined as: $RRI = 100 \times ((RR_{\text{obs}} - RR_{\text{rand}}/RR_{\text{rand}})$.

## Network dependencies and centralization

We represent network dependencies for each claim as a multilayer network (*Kivela et al., 2014*) $M = (V_{\text{M}}, E_{\text{M}}, L)$ (*Figure 3A*). In each network layer $L$, nodes $V$ are biomedical papers and edges $E$ between pairs of papers represent either a binary relationship of agreement ($L_1$) or the amount of overlap between the authors ($L_2$), methodologies ($L_3$), and references to prior publications ($L_4$) in the two papers. We quantify the amount of overlap between research papers using the Jaccard coefficient (*JC*). For any attribute – i.e., authors, methods, or references – $A_i$ and $A_j$, *JC* is the size of intersection divided by the size of the union: $JC(A_i, A_j) = \frac{|A_i \cap A_j|}{|A_i| + |A_j| - |A_i \cap A_j|}$. The resulting quantity represents the edge weight between a pair of articles in the

respective network layers of shared authors ($L_2$), methods ($L_3$), and references ($L_4$). Each drug-gene claim constitutes an undirected, multilayer network of papers connected via such weighted edges across layers (see *Figure 3A*).

We define an independence score *IND* as the proportion of maximum possible edges (*Wasserman and Faust, 1994*) in a network layer $E_{\max} = \frac{E}{n(n-1)/2}$ not present, $IND = \frac{E_{\max} - W}{E_{\max}}$, where $W$ is the sum over all weighted edges in a claim's respective layer of shared authors, methods, or references. Our independence scores can be viewed as the probability that any two randomly chosen findings about a claim are obtained by disconnected sets of authors (social independence $S_{\mathrm{ind}}$), methods (methodological independence $M_{\mathrm{ind}}$), and references (knowledge independence $K_{\mathrm{ind}}$), respectively. The independence scores approach one when most papers with findings in support of a claim share no common authors, methods, and references, and 0 when all papers share all of their authors, methods, and references, respectively (see Appendix 1).

To quantify the centralization of research communities $C$ for each claim, we employed the Gini coefficient. The Gini coefficient is used to measure the heterogeneity of distributions in social and information networks (*Kunegis and Preusse, 2012*). The coefficient ranges between 0 and 1. In the context of a bipartite author-article network (*Figure 3B–C*), the coefficient approaches 0 when all investigators author equal numbers of articles about a claim and increases to 0.3 and above (depending on the number of articles), when one investigator authors all articles and all others author only one. The Gini coefficient can be also represented as a percentage ranging from 0 to 100, as in *Figure 2D–E*. While other measures of network centralization are available (e.g., Freeman's centralization [*Freeman, 1978*]), the Gini coefficient, and the Lorenz curve on which it is based, is independent from the underlying degree distribution, making it suitable for comparisons among networks with different size and mean degree (*Kunegis and Preusse, 2012*; *Badham, 2013*).

### Replication prediction models

We fit univariate, multivariate, and interaction logistic regression models to estimate odds ratios, relative risk, and predicted probabilities of claim replication as a function of support in the literature $L_{\mathrm{supt}}$, social independence $S_{\mathrm{ind}}$, methodological independence $M_{\mathrm{ind}}$, knowledge independence $K_{\mathrm{ind}}$, centralization $C$, journal prominence $J$, and experimental variability $CV$. First, for exploratory purposes, we model each variable independently (see *Figure 4A*, *Figure 4—figure supplement 1*, and *Supplementary file 3*). Second, we model our variables simultaneously (see *Figure 4A*, *Figure 4—figure supplement 1*, and *Supplementary file 4*):

$$\mathrm{logit}\,\mathrm{P}(R=1) = \beta_0 + \beta_1 \times L_{\mathrm{supt}} + \beta_2 \times S_{\mathrm{ind}} + \beta_3 \times M_{\mathrm{ind}} + \beta_4 \times K_{\mathrm{ind}} + \beta_5 \times C + \beta_6 \times J + \beta_7 \times CV$$

Third, we estimate two interaction models to examine the effect of support in the literature on replication success as a function of social independence and centralization, respectively (see *Figure 4B–C* and *Figure 4—figure supplements 2,3*):

$$\mathrm{logit}\,\mathrm{P}(R=1) = \beta_0 + \beta_1 \times L_{\mathrm{supt}} \times S_{\mathrm{ind}} + \beta_2 \times CV$$
$$\beta_0 + \beta_1 \times L_{\mathrm{supt}} \times C + \beta_2 \times CV$$

To estimate and visualize the logistic regression models, we used the `glm()` function, specifying binomial distribution and the canonical logit link function, and the packages effects (*Fox, 2003*), sjPlot (*Lüdecke, 2019*), and ggplot2 (*Wickham, 2016*), all in R.

### Data availability

The data and computer code associated with this analysis are available on the Open Science Framework at https://osf.io/xmvda/.

### Acknowledgements

We thank A Belikov, D Centola, J Denrell, E Duede, JPA Ioannidis, M Lewis, T Stoeger and participants at the University of Chicago Booth School of Business Organizations and Markets Workshop and MIT Economic Sociology Working Group for helpful discussions and comments. We thank E Demir for advice on curated drug-gene interactions; R Melamed for advice on high-throughput experimental data; W Catino for computational help; M Kivelä for help with his Multilayer Networks Library for Python (Pymnet); and T Natoli (Broad Institute) for advice on NIH LINCS L1000 data. We used MEDLINE/PubMed and Web of Science data to build the co-paper networks: we thank Clarivate Analytics for supplying the Web of Science data and T Ando for computing the journal eigenfactor scores.

**Valentin Danchev** is in the Department of Sociology, University of Chicago, Chicago, and the Meta-Research Innovation Center at Stanford (METRICS), Stanford University, Stanford, United States
vdanchev@stanford.edu

https://orcid.org/0000-0002-7563-0168

**Andrey Rzhetsky** is in the Departments of Medicine and Human Genetics, and the Institute for Genomic and Systems Biology, University of Chicago, Chicago, United States

https://orcid.org/0000-0001-6959-7405

**James A Evans** is in the Department of Sociology, University of Chicago, Chicago, and the Santa Fe Institute, Santa Fe, United States

jevans@uchicago.edu

https://orcid.org/0000-0001-9838-0707

*Author contributions:* Valentin Danchev, Conceptualization, Data curation, Investigation, Visualization, Methodology, Writing—original draft, Writing—review and editing; Andrey Rzhetsky, Conceptualization, Funding acquisition, Project administration, Writing—review and editing; James A Evans, Conceptualization, Supervision, Funding acquisition, Investigation, Writing—original draft, Writing—review and editing

*Competing interests:* The authors declare that no competing interests exist.

**Funding**

| Funder | Grant reference number | Author |
|---|---|---|
| Defense Advanced Research Projects Agency | Big Mechanism 14145043 | Valentin Danchev Andrey Rzhetsky James A Evans |
| National Science Foundation | SciSIP 1158803 | Valentin Danchev James A Evans |
| Air Force Office of Scientific Research | FA9550-15-1-0162 | James A Evans |

The funders had no role in study design, data collection and interpretation, or the decision to submit the work for publication.

**Decision letter and Author response**

Decision letter https://doi.org/10.7554/eLife.43094.030
Author response https://doi.org/10.7554/eLife.43094.031

## Additional files

### Supplementary files

• Supplementary file 1. Data about the corpus of 51,292 drug-gene interaction claims.
DOI: https://doi.org/10.7554/eLife.43094.017

• Supplementary file 2. Data about the sub-corpus of 2493 drug-gene interaction claims for which there are two or more published findings.
DOI: https://doi.org/10.7554/eLife.43094.018

• Supplementary file 3. Logistic regression models with claim replication $R$ [Replicated = 1, Non-replicated = 0] as response variable and predictors modelled independently.
DOI: https://doi.org/10.7554/eLife.43094.019

• Supplementary file 4. Logistic regression models with claim replication $R$ [Replicated = 1, Non-replicated = 0] as response variable and predictors modelled simultaneously.
DOI: https://doi.org/10.7554/eLife.43094.020

### Data availability

The datasets generated or analysed during this study are included in the manuscript and supporting files and have been made available at OSF (http://dx.doi.org/10.17605/OSF.IO/XMVDA).

The following dataset was generated:

| Author(s) | Year | Dataset URL | Database and Identifier |
|---|---|---|---|
| Valentin Danchev, Andrey Rzhetsky, James A Evans | 2019 | http://dx.doi.org/10.17605/OSF.IO/XMVDA | Open Science Framework, 10.17605/OSF.IO/XMVDA |

The following previously published datasets were used:

| Author(s) | Year | Dataset URL | Database and Identifier |
|---|---|---|---|
| The Broad Institute | 2015 | https://www.ncbi.nlm.nih.gov/geo/query/acc.cgi?acc=GSE70138 | NCBI Gene Expression Omnibus, GSE70138 |
| MDI Biological Laboratory, NC State University | 2016 | http://ctdbase.org/downloads/ | Comparative Toxicogenomics Database, Chemical-gene interactions |

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

## Appendix 1

DOI: https://doi.org/10.7554/eLife.43094.021

### Data

#### High throughput drug-gene interactions

We used the Library of Integrated Network-based Cellular Signatures (LINCS) Phase I L1000 data set (*Subramanian et al., 2017*) to estimate claim replication. The data set measures the expression of 978 landmark genes treated with wide range of perturbagens across cell lines, time points, and dosage levels (concentration), resulting in approximately 1.3M profiles. The experimental profiles are aggregated to 473,647 signatures, as represented in Level five data we used to perform our analysis. The landmark genes are subsequently used to infer gene expressions for the remaining genes in the human genome. In addition to the 978 landmark genes, we consider 9196 for which the LINCS L1000 project estimated to be well inferred, resulting in 10,174 Best INferred Genes (BING) in total. With respect to perturbagen types, we used the set of small-molecule compounds (19,811 compounds), which includes a subset of approximately 1,300 FDA-approved drugs. We accessed the data file GSE70138_Broad_LINCS_Level5_COMPZ_n118050 × 12328_2017-03-06.gctx.gz and metadata from the GEO depository at https://www.ncbi.nlm.nih.gov/geo/query/acc.cgi?acc= GSE70138.

#### Published drug-gene interactions

We analysed curated data about published interactions between chemicals/drugs and genes/ mRNAs released by the Comparative Toxicogenomics Database (CTD) (*Davis et al., 2017*) on June 7 2016. (The current CTD data release is available here: http://ctdbase.org/downloads/). To align triples of drug, gene, and interaction effect in CTD to corresponding triples in the experimental signatures from LINCS L1000, we performed the following procedures. First, we selected drug-gene interactions about Homo sapiens, comprising approximately 40% of the CTD data. Second, CTD reports the form of the gene (e.g., mRNA, protein) that is implicated, and we selected only mRNA as LINCS L1000 measures gene expression at the mRNA level. Third, we mapped chemical names and Entrez gene IDs in the CTD to perturbagen names and Entrez gene IDs in LINCS L1000. Fourth, to ensure comparability to the LINCS L1000 signatures, we selected drug-gene interactions with a single interaction effect, either 'decreases expression' or 'increases expression', defining the direction of the effect that chemical/drug manifests on a gene/mRNA. Note that we do not consider complex interactions with multiple, nested effects in our analysis. Likewise, interactions for which the direction of the effect is not specified, such as 'affects binding', are not considered. The resulting corpus at the intersection of LINCS L1000 and CTD comprises 51,292 drug-gene claim combinations of 605 unique drugs and 9123 unique genes, annotated from 3363 scientific articles.

### Variables of claim provenance extracted from article metadata and content

#### Social independence and centralization

We used the MEDLINE/PubMed database to extract the set of authors for each paper. To measure the overlap between two sets of authors, we need individual author identifiers. Author name disambiguation is a common problem in research on scientific knowledge production. We used the individual identifiers based on author last name and initials. We note that because we assessed authors separately for each claim, our conservative matching procedure is very unlikely to produce false positive author linkages, and so our author co-paper network should be considered a lower bound for author co-paper density. For the sub-corpus of 2493 claims, sourced from 1282 papers, we estimated a mean of 6.4 authors per paper and a mean of 23.5 authors per scientific community defined here as the total number

of authors that have published papers reporting a drug-gene claim. For the set of papers supporting a claim, we used JC to measure the overlap between the authors in the various pairs of papers in the set, and then used our independence score to calculate the social independence $S_{ind}$ of the claim. Further, we applied the Gini coefficient to the bipartite author-article network for each claim to compute community centralization.

### Methodological independence

We compiled a controlled vocabulary of 3074 terms (incl. synonyms) concerning methods, techniques, and experimental apparatus used in biomedical research using ontologies of biomedical investigations (*Bandrowski et al., 2016*) and statistics (*Gonzalez-Beltran et al., 2016*). We then used the RESTful API Web Service of Europe PMC to query the methods sections from 4.4 million full text articles and extracted, on aggregate, 13,095 terms for 488 articles (38%). In parallel, for all 1282 articles that share a drug-gene claim, we applied fuzzy matching against our vocabulary using the difflib module in Python and extracted 12,135 terms from abstracts available in MEDLINE/PubMed. We combined outputs from the two search procedures. Then, for the set of papers supporting a claim, we again used JC to measure the overlap between methods in the various pairs of papers in the set, and then used our independence score to calculate the methodological independence $M_{ind}$ of the claim.

### Prior knowledge independence

To examine whether a pair of publications is exposed to similar or dissimilar prior information, we use the notion of bibliographic coupling (*Kessler, 1963*), i.e., the number of citations any two papers share. To compute bibliographic coupling, we used the Web of Science citation data. Out of 1282 papers sharing a drug-gene claim with at least one other paper, we mapped 1234 PubMed IDs to Web of Science IDs and performed bibliographic coupling on this subset using Python modules Tethne (*Peirson and Erick, 2017*) and NetworkX (*Hagberg et al., 2008*). 880 of our 1234 papers were coupled bibliographically by at least one paper. Consistent with the procedure we applied to measure shared authors and methods, we used JC to measure the overlap between the citations in the various pairs of papers in the set, and then used our independence score to calculate the prior knowledge independence $K_{ind}$ of the claim.

### Journal prominence

To measure journal prominence, we employed the journal eigenfactor score (*Bergstrom et al., 2008*). The eigenfactor score acts like a recursively weighted degree index by rating journals highly that receive citations from journals that are themselves highly cited. Journal eigenfactor scores were computed using the Web of Science database. We obtained journal eigenfactor scores for 3162 papers (94% of all 3363 papers in our corpus) published in 656 journals between 1995 and 2016. For the sub-corpus of 1282 papers with shared drug-gene claims, we recovered 1212 papers or 95% published in 496 journals. For each claim, we computed mean journal eigenfactor scores by averaging over the eigenfactor score of all journals that published a paper reporting findings in support of the claim. The distribution of mean journal eigenfactor scores (i.e., journal prominence) per claim is highly skewed (*Figure 1—figure supplement 2* and *Figure 4—figure supplement 4*), indicating that claims receive overwhelmingly support from findings published in low and medium ranked journals. This highlights the value of archiving findings across a wide range of journals, as do CTD and other scientific database projects, which makes possible our large-scale evaluation of scientific output.

## Robustness analysis

Some of our variables are correlated (e.g., methodological independence and prior knowledge independence; see *Figure 3—figure supplement 1*), which is to be expected as they capture related dimensions of scientific knowledge production. We performed a multicollinearity test using the variance inflation factor (VIF). The variance inflation factors vary from low for variability in LINCS L1000 (VIF = 1.015), journal prominence (VIF = 1.135), and support in literature (VIF = 1.534) to moderate for centralization (VIF = 3.162), methodological independence (VIF = 3.752), prior knowledge independence (VIF = 4.681), and social independence (VIF = 4.900). We observe no predictor with high variance inflation factor, i.e., VIF $\geq$ 10. We removed the two variables with the highest VIF >4 and refit our logistic regression model. In the refitted model, both support in the literature and decentralization of scientific communities remain strong and significant predictors of replication success (*Figure 4—figure supplement 5*). Further, we verified that the effects of support from the literature and community centralization on claim replication success are not dominated by outliers. Recall the long-tailed distribution of findings per claim, with few claims receiving support from many published findings (*Figure 1A*). We removed claims supported by 10 or more findings, amounting to 26 claims supported by 426 findings and found that support from the literature (OR 34.785; 95% CI, 12.518–96.662, p = 1.00e-11) and community centralization (OR 0.335; 95% CI, 0.249–0.451, p = 5.57e-13) remain strong and significant predictors of replication success, after adjusting for biological and experimental variability in LINCS L1000 (OR 0.726; 95% CI, 0.429–1.232; p = 0.236). Similarly, the distribution of findings per paper and per pair of papers is heterogeneous (*Figure 3—figure supplement 2*). We removed the largest set of 796 drug-gene claims reported by a pair of papers and found that the effect of support from the literature (OR 19.512; 95% CI, 7.193–52.929, p = 5.37e-09) and centralization (OR 0.432; 95% CI, 0.287–0.65, p = 5.71e-05) holds and is not explained by variability in LINCS L1000 (OR 0.574; 95% CI, 0.279–1.181; p = 0.131).

