## [Decision Letter]

Thank you for submitting your article "Centralized Scientific Communities Less Likely to Generate Replicable Results" to *eLife*. Your article has been reviewed by three peer reviewers, and the evaluation has been overseen by Peter Rodgers, the *eLife* Features Editor. The following individual involved in review of your submission have agreed to reveal his identity: Luis AN Amaral (Reviewer #1).

The reviewers have discussed the reviews with one another and the *eLife* Features Editor has drafted this decision to help you prepare a revised submission.

Summary:

Danchev, Rzhetshy, and Evans investigate the replicability and reliability of published findings by comparing published findings with the high throughput results from the NIH LINCS L1000 program. This is a clever approach. The paper is overall well written, and the analysis is thorough and complete. All in all, the authors provide important results that may help guide the development of more reliable science.

Essential revisions:

1) *eLife* papers use the Introduction/Results/Discussion/Materials and methods format, so you will need to reorder these sections in your manuscript. Also, *eLife* papers do not include supplementary information: rather, figures in the supplementary information need to become supplements to the figures in the main text (called Figure 1—figure supplement 1 etc.), and the text in the supplementary information needs to be moved to the main text (usually to the methods section).

2) The authors' analysis seems to rely on assuming that the LINCS L1000 data is true. A brief discussion about the importance of this assumption and perhaps what would happen were it violated would be helpful.

3) While this study is very thorough as it stands, I believe that the authors have the opportunity to answer another question that could increase the impact of the paper. The authors find that most DGI claims are only reported in a single study. Why is this? I can imagine two scenarios: i) Gene is not thought to be important, so there is no incentive for further study, or ii) Gene is important but claim is by star scientist and attempts at publishing contradictory claim are stymied. It would be interesting to investigate whether the data support either of these scenarios (or another scenario). If the authors feel that such an analysis is beyond the scope of the present paper, they should at least comment on this issue.

4) The reader needs more detail about bootstrapping (what is being bootstrapped? What is the typical sample size? What is the distribution of perturbations for each gene?). Please either add these details to the subsections “High-throughput Claim Replication Strategy”, last paragraph, “Experimental Generalizability and Variability” and “Measuring Relative Replication Increase”, or include references to the relevant parts of the methods section.

5) Please clarify how replication is defined. What if LINCS data is not consistent? Or if the other data/papers are not consistent (e.g. includes + and – results)?

6) In the first paragraph of the Discussion, the authors imply a causal relationship between centralized communities and reliability of scientific findings. I recommend that this (and any other similar statements) be modified so as to not imply causation. It could be that some third confounding factor leads to lower reliability. For instance, this sentence could be written: "Our results indicate that such communities are associated with lower reliability of scientific findings." If the authors believe this analysis can discern causality, please describe why.

7) Figure 1. Panel A would make more sense to me as a scatter plot of number of supporting claims versus number of opposing claims. Panel C would be easier to read if the survival function was plotted instead of a frequency plot. In their current forms Panels B and D containing relatively little information: please consider deleting these panels and including the relevant data in the figure legend. Also, regarding panel D: it would be more interesting to plot the fractions that are significant as the p-value threshold for significance is changed.

8) Figure 2. It is not clear to me what subset of data is being considered in Panel B. It appears to be different from the subset in Panel C since the number of contradictory claims is different for the two panels (43 vs. 77). I would name the different subsets of data, provide a table with their definition, and use the names in the text so as to help the reader. Also, please explain why there are two random groups in Panel C. Also, 'contradictory' seems to me not a good name since a potentially large majority of claims does not support the existence of the interaction: 'not supported' would appear to be a better designation.

9) In figures including p-values (e.g. Figure 2D and 2E), it would be helpful to have p-values rather than stating "n.s." Additionally, clarity on the p-values and how they are calculated in Figure 2D and 2E would be helpful

10) Figure 3. Please provide a y-axis label for Panel B. This panel is also difficult to read so please consider having the lines end at the middle of the bottom edge of the rectangle/square, rather than inside the rectangle/square.

11) Figure 4A claims to show the OR for replication in LINCS L1000 data but the figure legend claims that something else is being plotted. Moreover, several studies have showed that OR are usually misinterpreted and that use of relative risk (RR, which can be derived from the OR) is more informative of the actual impact of a factor. I recommend also plotting the RR.

12) Figure 4—figure supplement 3C: the 95% confidence intervals for the top 2 rows do not appear to contain any of the parameters in the table. More clarification on what they represent is needed.

---

## [Author Response]

Essential revisions:1) eLife papers use the Introduction/Results/Discussion/Materials and methods format, so you will need to reorder these sections in your manuscript. Also, eLife papers do not include supplementary information: rather, figures in the supplementary information need to become supplements to the figures in the main text (called Figure 1—figure supplement 1 etc.), and the text in the supplementary information needs to be moved to the main text (usually to the Materials and methods section).

We have reordered the manuscript accordingly and incorporated supplementary figures and information in the main text and in Appendices.

2) The authors' analysis seems to rely on assuming that the LINCS L1000 data is true. A brief discussion about the importance of this assumption and perhaps what would happen were it violated would be helpful.

While we evaluate the reliability and biases in published DGI claims against the LINCS L1000 experiments, our approach does not rely on the LINCS L1000 data being completely correct or true. Rather, we argue that the LINCS L1000 data are (1) produced with methods representative of research in this area, and (2) free from the social, prior knowledge and selective technical dependencies associated with the sequential publication of findings by communities of researchers. In this way, our manuscript attempts to elucidate biases introduced by the social, cultural and technical structure of science to noisy experimental data. In this framework, LINCS L1000 experiments must remain unfiltered by the publication system and expectations of any particular scientific community. To clarify this, we have these details to the current Discussion section.

3) While this study is very thorough as it stands, I believe that the authors have the opportunity to answer another question that could increase the impact of the paper. The authors find that most DGI claims are only reported in a single study. Why is this? I can imagine two scenarios: i) Gene is not thought to be important, so there is no incentive for further study, or ii) Gene is important but claim is by star scientist and attempts at publishing contradictory claim are stymied. It would be interesting to investigate whether the data support either of these scenarios (or another scenario). If the authors feel that such an analysis is beyond the scope of the present paper, they should at least comment on this issue.

In response to this thoughtful suggestion, we have explored reasons behind the tendency for most DGI claims to be reported in a single paper. Because our research design is tailored towards the examination of network dependencies underlying biomedical claims reported in multiple published studies, our analysis of single-studied claims is limited and of an exploratory nature.

First, a major factor associated with single-studied claims in our corpus appears to be their relatively low replication likelihood, indicating that many could have been obtained by chance, minimizing the possibility of further reporting. Consider our finding (Figures 2D and 2E) that DGI claims studied in a single study (“Low” support) are less likely replicated in LINCS L1000 than DGI claims reported in multiple studies (“Very high” and “High” support). Further, focusing only on the first published study among multiple-studied claims with “Very high” and “High” support, we find that those also have higher observed replication rate of 0.7196 than single-studied claims in our corpus (“Low” support), which yield an observed replication rate of 0.6010. (Because opposing findings are rare, the replication rates for the first published finding and for all published findings about multiple-studied claims (“Very high” and “High” support) are very similar: 0.7196 and 0.7232, respectively.) In the event that a single-studied claim was tested again, researchers would be less likely to obtain positive results compared to a multiple-studied claim, placing the finding in the “file drawer”. We relate this to the expectation of novel results in the current publication system that drives the introduction of novel but not necessarily reliable claims, less likely to be confirmed in subsequent studies.

Second, among all claims in our corpus, we find no significant correlation between the number of findings published about a claim and the importance of the publication journal measured using the eigenfactor score [1] (Pearson corr. coef. = -0.0026, *P* = 0.5614, n = 50,346; journal eigenfactor scores are available for 50,346 of 51,292 claims).

Third, we found that DGI claims contributed from one or more researchers who published five or more articles in the whole corpus are much more likely to be reported in multiple studies (17.7%) than claims contributed only by researchers who published less than five articles in our corpus (4.1%). Recall in the manuscript that we found centralization of scientific communities correlates with overlapping methods, see Figure 3—figure supplement 1. Consistent with that finding, multiple studies from a related set of researchers tended to cluster on a few claims. Conversely, authors that publish a small number of articles in our corpus rarely add to the network of existing claims, but tend to introduce isolated, “singleton” claims.

Fourth, we examined whether centralized sources of a claim would discourage outside researchers from reporting findings about that claim, as suggested by the reviewer. We add the following to the manuscript: “Social dependencies could impact the composition of researchers studying a claim by discouraging researchers outside the dense communities of researchers who initially lay claim to it from pursuing or reporting their findings [2]. Focusing on “Very high” and “High” support claims with some degree of centralization (n = 295), we found that claims that were both originally reported and subsequently repeated and confirmed in by papers with overlapping authors (n = 117) ended up as much more centralized communities (mean C = 0.6164; 95CI, 0.5925 to 0.6402) compared to claims reported and subsequently repeated and confirmed by papers with independent authors (n = 178; mean C = 0.4004; 95CI, 0.3765 to 0.4242; two-way ANOVA test with unbalanced design.) This exploratory result suggests that if claims exhibit social dependencies very early in their development, they likely receive limited attention and reportage from researchers outside the overlapping communities that originated them.” This is broadly consistent with the finding by Azoulay and colleagues, cited within this section, that demonstrates how closed, high-status communities in biomedicine not only discourage, but also block participation from others.

We have added elements from each of these demonstrations into the manuscript.

4) The reader needs more detail about bootstrapping (what is being bootstrapped? What is the typical sample size? What is the distribution of perturbations for each gene?). Please either add these details to the subsections “High-throughput Claim Replication Strategy”, last paragraph, “Experimental Generalizability and Variability” and “Measuring Relative Replication Increase”, or include references to the relevant parts of the Materials and methods section.

We agree and have added details about the bootstrapping procedure and substantially revised text at the specified lines. We provide one example for reference:

“We bootstrapped (10,000 iterations) each sample of moderated z-scores per DGI to estimate confidence intervals. The bootstrapped samples vary across DGIs as a function of the number of cell lines, dosages, and durations under which the DGI was tested (the mean and median of moderated z-scores per DGI in our corpus are 143.8 and 49, respectively).”

5) Please clarify how replication is defined. What if LINCS data is not consistent? Or if the other data/papers are not consistent (e.g. includes + and – results)?

Our definition of claim replication triangulates the following components. First, we consider the amount of support a claim received from the literature and categorized claims into five classes on the basis of their respective amount of support. In the literature, if both positive and negative results are reported about a claim, the claim would be categorized as “Low” support or “Contradictory”, depending on the distribution of positive and negative findings. In LINCS L1000, inconsistent interactions would not, by definition, generalize across conditions, and, on this basis, will not be included in the central analysis. Then we consider if DGIs in the LINCS L1000 data match the direction of the effect in the literature. We define replication as a binary variableR∈{0,1}, where *R* = 1 indicates that LINCS L1000 matches the direction (increase or decrease) of the effect in literature, and *R* = 0 otherwise. In summary, to account for cases in which the LINCS L1000 and/or the literature are inconsistent, our claim replication strategy keeps track of (1) the amount of support in the literature and (2) the generalizability of interactions in LINCS L1000 when replicability of published DGI claims is evaluated against LINCS L1000 high-throughput experiments.

6) In the first paragraph of the Discussion, the authors imply a causal relationship between centralized communities and reliability of scientific findings. I recommend that this (and any other similar statements) be modified so as to not imply causation. It could be that some third confounding factor leads to lower reliability. For instance, this sentence could be written: "Our results indicate that such communities are associated with lower reliability of scientific findings." If the authors believe this analysis can discern causality, please describe why.

We agree with the reviewers’ concern and we rewrote this and similar statements to reflect the fact that our research design identifies associations rather than causal relationships. In addition to rewriting the above sentence as suggested by the reviewer, we rewrote the sentence which now reads: “Decentralized, sparsely connected communities explore diverse research methodologies and prior knowledge (Figure 3—figure supplement 1), and are more likely to generate replicable findings.” We use this association to propose a causal hypothesis, but we do not claim that we have causally identified a finding.

7) Figure 1. Panel A would make more sense to me as a scatter plot of number of supporting claims versus number of opposing claims. Panel C would be easier to read if the survival function was plotted instead of a frequency plot. In their current forms Panels B and D containing relatively little information: please consider deleting these panels and including the relevant data in the figure legend. Also, regarding panel D: it would be more interesting to plot the fractions that are significant as the p-value threshold for significance is changed.

We implemented most of these suggestions. As recommended, we plotted the survival probability in Panel C. In addition, we removed Panels B and D. Panel B is currently included as an inset figure in Panel A. The information conveyed in Panel D is currently plotted in Panel B.

We considered plotting the number of supporting findings versus the number of opposing findings but because the majority of findings in the literature are supporting, visualizing the two quantities poses a challenge.

8) Figure 2. It is not clear to me what subset of data is being considered in Panel B. It appears to be different from the subset in Panel C since the number of contradictory claims is different for the two panels (43 vs. 77). I would name the different subsets of data, provide a table with their definition, and use the names in the text so as to help the reader. Also, please explain why there are two random groups in Panel C. Also, 'contradictory' seems to me not a good name since a potentially large majority of claims does not support the existence of the interaction: 'not supported' would appear to be a better designation.

We regret that this was unclear. We have now clarified the two subsets of data we used in Figure 2B and C, as suggested. We have also included a table that clarifies the subsets of data as well as the number of claims pertaining to each subset. With regard to the two random groups in Panel C, they correspond to claims that are found to be significant and non-significant in LINCS L1000, respectively. We agree with the reviewer that the notion “contradictory” might not convey the actual distribution of support for those claims. We have adopted the suggested name “not supported” for those claims.

9) In figures including p-values (e.g. Figure 2D and 2E), it would be helpful to have p-values rather than stating "n.s." Additionally, clarity on the p-values and how they are calculated in Figure 2D and 2E would be helpful

We have added the actual p-values in the two figures and clarified the approach we used to calculate them.

“We bootstrapped 100,000 times the *RRI* distributions for each class of claims and calculated the p-value as the proportion of iterations in which the difference between two claim classes’ *RRI* distributions contains 0.”

10) Figure 3. Please provide a y-axis label for Panel B. This panel is also difficult to read so please consider having the lines end at the middle of the bottom edge of the rectangle/square, rather than inside the rectangle/square.

We thank for those helpful suggestions. We have added a y-axis and improved the readability for Panel B in Figure 3, as suggested.

11) Figure 4A claims to show the OR for replication in LINCS L1000 data but the figure legend claims that something else is being plotted. Moreover, several studies have showed that OR are usually misinterpreted and that use of relative risk (RR, which can be derived from the OR) is more informative of the actual impact of a factor. I recommend also plotting the RR.

We have clarified in the caption of Figure 4A that we show odds ratios derived from logistic regression models [with replication as a binary response variable and predictors modeled independently (dots) and simultaneously (triangles)]. Specifically, we fitted a set of generalized linear models using the glm() function in R, specifying binomial distribution and the canonical logit link function. We then took the exponent of the coefficients (log odds) to obtain the odds ratios we show in Figure 4A. As suggested by the reviewer, we have also derived the relative risk (RR) from our logistic models (see Figure 4A) using the following method [3] and the sjstats [4] package for R. (We also considered alternative specifications to directly estimate relative risk using poisson / quasipoisson regression, typically used for count outcomes, but a reliable poisson / quasipoisson regression for binary outcomes would involve additional assumptions and modifications [5].). We integrated in Figure 4—figure supplement 1 the plot of RR of claim replication with plots of predicted probabilities (PP) of claim replication derived from the independent logistic models in Figure 4A.

12) Figure 4—figure supplement 3C: the 95% confidence intervals for the top 2 rows do not appear to contain any of the parameters in the table. More clarification on what they represent is needed.

Thank you for pointing out this problem. We have identified the source of the misalignment. While the coefficients (and the 95% confidence intervals) were correct, the exponentiated values did not refer to the correct coefficients. We have exponentiated the correct coefficients and included the resulting values in the table.

References:

1) Bergstrom CT, West JD, Wiseman MA. The Eigenfactor Metrics. The Journal of Neuroscience 2008;28:11433.

82) Azoulay P, Fons-Rosen C, Zivin JSG. Does Science Advance One Funeral at a Time? National Bureau of Economic Research Working Paper Series 2015;No. 21788.

3) Zhang J, Yu KF. What's the Relative Risk?A Method of Correcting the Odds Ratio in Cohort Studies of Common Outcomes. JAMA 1998;280:1690-1.

4) Lüdecke D. sjstats: Statistical Functions for Regression Models. 2019; R package version 0.17.5, doi: 10.5281/zenodo.1284472, https://CRAN.R-project.org/package= sjstats..

5) Zou G. A Modified Poisson Regression Approach to Prospective Studies with Binary Data. American Journal of Epidemiology 2004;159:702-6.